# Large quantum-spin-Hall gap in single-layer 1$T'$ WSe$_2$

P. Chen [1,2,3], Woei Wu Pai[4,5,6], Y.-H. Chan[7], W.-L. Sun[8], C.-Z. Xu [1,2], D.-S. Lin[8], M.Y. Chou[5,6,9], A.-V. Fedorov[3] & T.-C. Chiang[1,2,5]

Two-dimensional (2D) topological insulators (TIs) are promising platforms for low-dissipation spintronic devices based on the quantum-spin-Hall (QSH) effect, but experimental realization of such systems with a large band gap suitable for room-temperature applications has proven difficult. Here, we report the successful growth on bilayer graphene of a quasi-freestanding WSe$_2$ single layer with the 1$T'$ structure that does not exist in the bulk form of WSe$_2$. Using angle-resolved photoemission spectroscopy (ARPES) and scanning tunneling microscopy/spectroscopy (STM/STS), we observe a gap of 129 meV in the 1$T'$ layer and an in-gap edge state located near the layer boundary. The system's 2D TI characters are confirmed by first-principles calculations. The observed gap diminishes with doping by Rb adsorption, ultimately leading to an insulator–semimetal transition. The discovery of this large-gap 2D TI with a tunable band gap opens up opportunities for developing advanced nanoscale systems and quantum devices.

[1] Department of Physics, University of Illinois at Urbana-Champaign, 1110 West Green Street, Urbana, IL 61801-3080, USA. [2] Frederick Seitz Materials Research Laboratory, University of Illinois at Urbana-Champaign, 104 South Goodwin Avenue, Urbana, IL 61801-2902, USA. [3] Advanced Light Source, Lawrence Berkeley National Laboratory, Berkeley, CA 94720, USA. [4] Center for Condensed Matter Sciences, National Taiwan University, Taipei 10617, Taiwan. [5] Department of Physics, National Taiwan University, Taipei 10617, Taiwan. [6] Center of Atomic Initiative for New Materials, National Taiwan University, Taipei 6 10617, Taiwan. [7] Institute of Atomic and Molecular Sciences, Academia Sinica, Taipei 10617, Taiwan. [8] Department of Physics, National Tsing Hua University, Hsinchu 30013, Taiwan. [9] School of Physics, Georgia Institute of Technology, Atlanta, GA 30332, USA. Correspondence and requests for materials should be addressed to P.C. (email: pchen229@illinois.edu) or to T.-C.C. (email: tcchiang@illinois.edu)

The quantum-spin-Hall (QSH) insulator films are characterized by a two-dimensional (2D) band gap within the film and one-dimensional (1D) metallic edge states that bridge the band gap[1–5]. The edge states are spin polarized by spin–orbit coupling in a chiral configuration relative to the momentum and the edge normal; they are protected by time-reversal symmetry and thus robust against weak disorder. These edge conducting channels are ideally suited for low-dissipation transport of spin information relevant to spintronic applications[3,4]. The first experimental demonstration of the QSH effect was made in HgTe/(Hg, Cd)Te quantum wells[6,7], but the system configuration was complex; furthermore, its gap was very small, and the edge spin conduction effect was observed only at very low temperatures. Nevertheless, the proof of principle has spurred a great deal of community interest in finding simple robust 2D TI systems with a large band gap. Extensive theoretical explorations have been made in various systems ranging from 2D elemental buckled lattices to transition-metal pentatellurides and to oxide heterostructures[8–15], but experimental realization of systems with properties readily amenable to applications has proven to be elusive.

Transition-metal dichalcogenides are promising for developing 2D topological insulators (TIs). These layered materials can be easily fabricated in single-layer forms. Some of them containing heavy elements with a strong spin–orbit coupling are predicted to be 2D TIs; notable examples include single-layer $1T'$ $MX_2$ (M = Mo or W; X = S, Se, or Te)[16]. Experimental work to date has mostly focused on single-layer $WTe_2$ because the $1T'$ phase is the stable form in the bulk[17–19], but not necessarily so for the other cases, and W has a very strong atomic spin–orbit coupling. While some of the other $MX_2$ materials can be converted to the $1T'$ phase by intercalation or strain, the added complexity makes it difficult to prove the QSH state[20]. $WSe_2$, the material chosen for the present study, exhibits a hexagonal $2H$ structure in the bulk, which is of great interest for its large indirect band gap and strong spin-valley coupling[21,22], but the single layer with the $1H$ structure is not topological.

In this work, we show that single layers of $WSe_2$ can be prepared instead in the $1T'$ phase, which is topological with a gap of 129 meV based on angle-resolved photoemission spectroscopy (ARPES) experiments and 116 meV based on $G_0W_0$ calculations. This gap is more than twice as large as that reported for $1T'$ $WTe_2$[17] and, furthermore, it can be tuned with surface doping to undergo an insulator–semimetal transition. Single-layer $1T'$ $WSe_2$ is thus an excellent candidate for developing spintronics based on the QSH effect.

## Results

**Film structure and electron diffraction patterns.** Figure 1a shows top and side views of the structure of $1T'$ and $1H$ single-layer $WSe_2$. The observed bulk crystal structure is the $2H$ phase consisting of a van der Waals stack of $1H$ layers in an ABA sequence. The bulk $1T'$ structure, if existent (as in $WTe_2$), would involve an ABC stacking of the $1T'$ layers together with a lattice distortion along the $x$ direction. The surface unit cells and some special points are indicated in Fig. 1b. In our experiment, films of $WSe_2$ were grown in situ on a bilayer graphene-terminated 6H-SiC(0001)[23] via van der Waals epitaxy[24,25]. Reflection high energy electron diffraction (RHEED) shows that a single-layer $WSe_2$ grown at a substrate temperature of 280 °C (Fig. 1c) exhibits a mixture of $1H$ and $1T'$ phases, both with sharp diffraction patterns. At higher substrate growth temperatures, the $1H$ phase becomes more prevalent, and it is the only phase observed at growth temperatures above 400 °C (Fig. 1c). The $1T'$ phase is favored

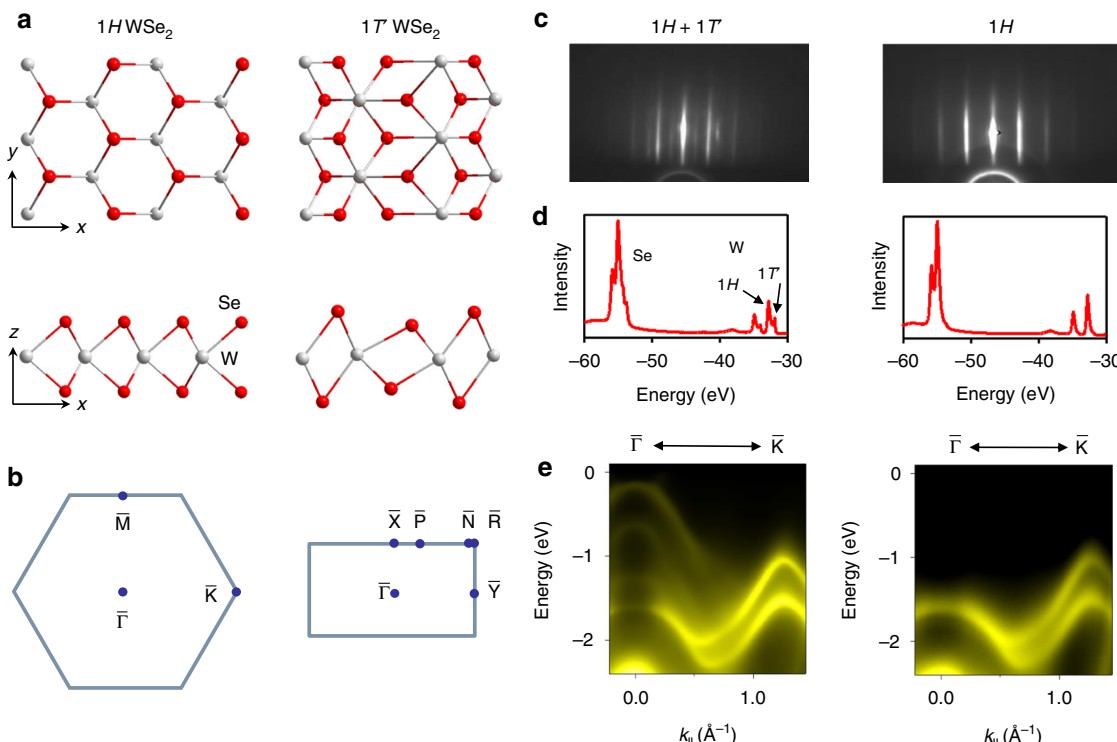

**Fig. 1** Film structure and electronic band structure of single-layer $WSe_2$. **a** Top and side views of the atomic structure of single-layer $1H$ and $1T'$ $WSe_2$. **b** Corresponding 2D Brillouin zones with high symmetry points labeled. **c** RHEED patterns taken from a $1H+1T'$ sample and a pure $1H$ sample. **d** Core level scans taken with 100 eV photons. The $1H+1T'$ sample shows mixed core level signals. **e** ARPES maps along $\overline{\Gamma K}$ taken from the two samples at 10 K

at lower growth temperatures, and it becomes the only phase observed at a growth temperature of 130 °C; however, the film quality is poor as evidenced by a fuzzy RHEED pattern (not shown here). Scans of the core levels (Fig. 1d) show splittings of the Se 3d and W 4f states in the mixed phase due to the inequivalent structures; an analysis of the W core level line shape indicates that the 1H and 1T′ phases have a coverage ratio of 1.8 on the surface.

**Band gap determined from ARPES and calculation.** ARPES maps taken from the single-layer samples at 10 K along the $\overline{\Gamma K}$ direction are shown in Fig. 1e. The pure 1H phase shows a sizable gap below the Fermi level; it is therefore a semiconductor similar to the bulk case[21]. The top valence band at $\overline{\Gamma}$ splits into two branches toward $\overline{K}$ because of the strong spin–orbit coupling of W. The valence band maximum is at $\overline{K}$, consistent with prior studies of this phase[24,26]. For the mixed sample, the 1T′ phase gives rise to additional valence bands of very different dispersion relations, and the topmost valence band reaches near the Fermi level to form a fairly flat portion around the zone center. For comparison, calculated band structure of the 1T′ phase based on the generalized gradient approximation/Perdew–Burke–Ernzerhof (GGA/PBE) method is presented in Fig. 2b. This phase is also a semiconductor but with an indirect band gap along the $\overline{\Gamma Y}$ direction. A band inversion occurs at $\overline{\Gamma}$ between the W 5d and Se 4p orbitals with opposite parities; spin–orbit coupling causes anti-crossing, giving rise to a gap of $E_g = 33$ meV based on the calculation (Fig. 2c). The GGA/PBE scheme tends to underestimate semiconductor gaps; a $G_0W_0$ calculation yields a more accurate gap value of 116 meV. With an inverted band

topology across a spin–orbit gap, the 1T′ phase is expected to be a 2D TI. Indeed, our computed topological index $Z_2$ equals 1, confirming that the system is a QSH phase (but not for the 1H phase which is an ordinary insulator).

The 1T′ phase has a rectangular lattice (Fig. 1a) instead of a triangular lattice; as a result, it exhibits three domains related by 120° rotations. ARPES spectra taken along $\overline{\Gamma Y}$ and $\overline{\Gamma X}$ include contributions from $\overline{\Gamma P}$ and $\overline{\Gamma N}$, respectively (Fig. 2a). Detailed ARPES maps near the Fermi level for the 1T′ phase together with overlaid theoretical band structure along the different directions (Fig. 2d) confirm the mixed-domain configuration. Specifically, the top valence band shows an apparent "splitting" near 0.2 Å$^{-1}$, which is a consequence of the domain mixing. The conduction band minimum is clearly seen in Fig. 2d. As indicated, the indirect gap between the valence band maximum and the conduction band minimum is 129 meV, which is very close to the $G_0W_0$ value of 116 meV. Notably, it is twice as large as that of the related 2D TI WTe₂[17]. The thermal energy $k_BT$ is 25 meV at room temperature, and semiconductors must have a gap greater than about $4k_BT = 100$ meV in order to be useful. Based on this criterion, 1T′ WSe₂ would be an excellent candidate for QSH applications at ambient temperature.

**Tunable band gap by surface doping.** For better viewing of the conduction bands, we have employed Rb doping to shift the bands downward relative to the Fermi level (Fig. 3a). Interestingly, the changes at higher doping levels cannot be described by a rigid shift of the bands. Specifically, the gap becomes smaller and the conduction and valence band edges eventually cross each

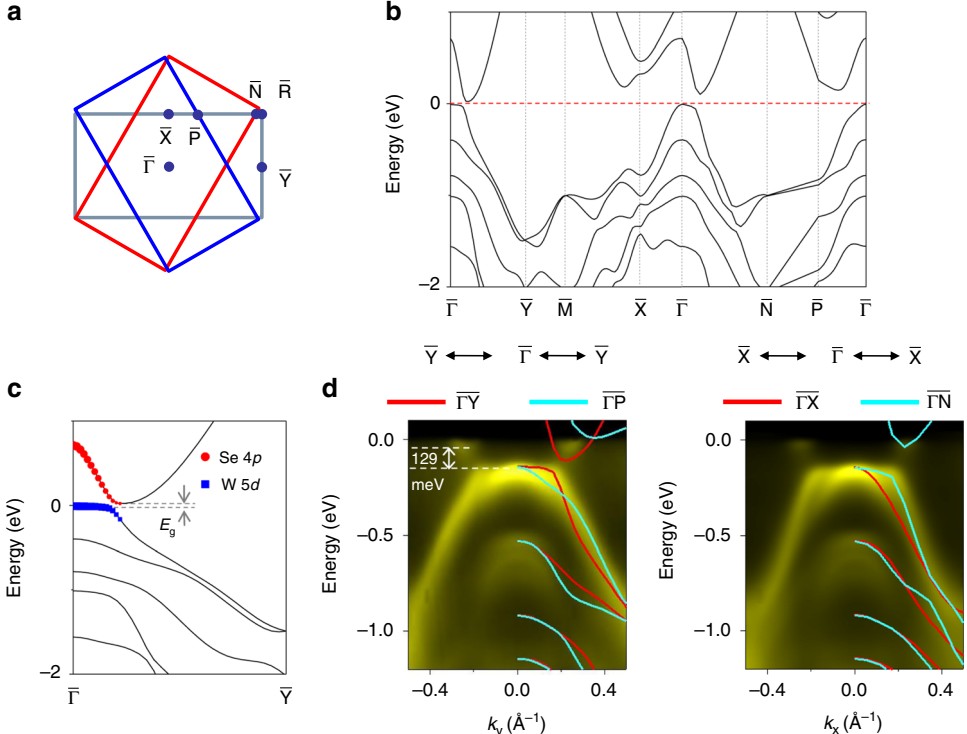

**Fig. 2** Band structure and band gap of 1T′ WSe₂. **a,** Brillouin zones of 1T′ WSe₂ with three domains separated by 120°. **b** Calculated band structure of 1T′ WSe₂. **c** Detailed band structure along $\overline{\Gamma Y}$ with the indirect gap $E_g$ labeled. The Se 4p and W 5d weights for the two topmost bands near the zone center are indicated by the red and blue dot sizes, respectively. **d** Two ARPES maps taken along $\overline{\Gamma Y}$ and $\overline{\Gamma X}$ with the sample at 10 K. The overlaid red and cyan curves are computed bands for the mixed-domain configurations. The experimental $E_g$ is indicated

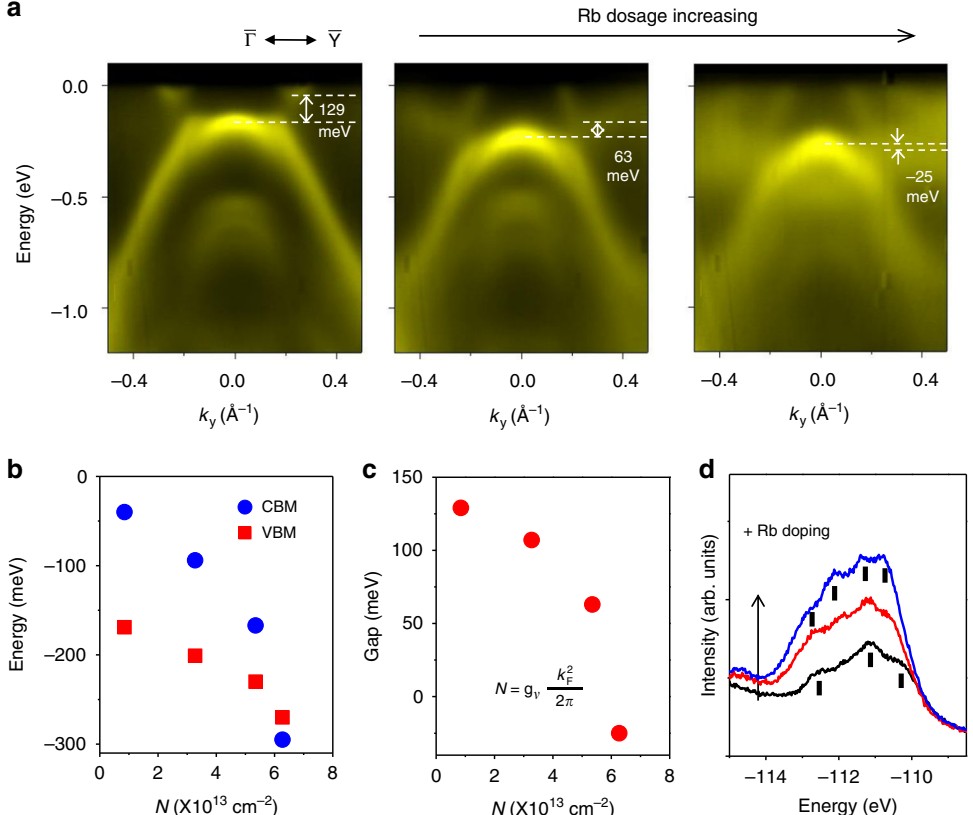

**Fig. 3** Tunable band gap in 1T' WSe$_2$ with Rb doping. **a** ARPES maps taken at 10 K for 1T' WSe$_2$ along the $\overline{\Gamma Y}$ direction with increasing Rb dosage. **b** Conduction band minimum (CBM) and valence band maximum (VBM) as a function of surface electron density. **c** Extracted band gap as a function of surface electron density. **d** Evolution of the Rb 3d core level line shape taken with 167 eV photons for increasing amounts of Rb dosage

other (Fig. 3b) at a doping level of $N_C = 6 \times 10^{13}$/cm$^2$, beyond which the system becomes a semimetal with a negative gap (Fig. 3c). Moreover, the band shapes become noticeably different. An implication is that Rb deposition leads to, in addition to surface electron doping, structural modifications through incorporation or intercalation of Rb in the lattice[22,27]. The Rb 3d core level line shapes (Fig. 3d) reveal multiple components indicating different Rb sites that vary in population with increasing Rb coverages. The tunability of the QSH gap can be a useful feature relevant to applications. The insulator–semimetal transition at $N_C$ offers a mechanism to switch off the QSH channels.

**Band gap and edge conductance measured by STM/STS**. The scanning tunneling microscopy/spectroscopy (STM/STS) measurements in a different chamber performed on a sample with a coverage of 1/2 layer made in the ARPES system transferred under a capping layer through air (see Methods section) reveal single-layer islands of 1H and 1T' structures and some two-layer islands (Fig. 4a). STS scans reveal a large band gap for the interior of 1H islands (Fig. 4d) and a much smaller gap for the interior of 1T' islands in agreement with the ARPES data, although the STS data are expected to be thermally broadened at the measurement temperature of 77 K relative to the ARPES data taken at 10 K. Figure 4b, c shows atom-resolved images of the 1T' and 1H phases, respectively. The orientation of the triangular 1H phase is the same as the underlying bilayer graphene, and the image exhibits a Moiré

pattern[24]. The 1T' phase has a rectangular lattice instead (Fig. 1a); all three domain orientations separated by 120° are observed.

Figure 4e is an STM image with a 1T' island covering the right half only. Figure 4f shows STS curves taken at a point (A) in the island very close to the island edge (green curve A) and another point (B) in the island still near but farther away from the edge (blue curve B); the two points A and B are indicted by the correspondingly color-coded dots in Fig. 4e. With the Fermi level aligned relative to the included ARPES map[28], the gap is indicated by the two vertical red dashed lines. Also shown is a reference red curve C taken from a point deep inside the island. It shows a clear gap; the small residual tunneling density of states (DOS) within the gap can be attributed to tunneling/coupling to the underlying bilayer graphene. Similar nonzero DOS in the gap is evident in single-layer WTe$_2$ results[17]. Curve A, relative to curve C, shows much higher DOS within the gap, suggesting contributions from edge states[13,14,17]; this extra DOS is much reduced for curve B. Details regarding the edge-state contributions through the island boundary are shown in Fig. 4g, where the differential conductance is plotted as a function of $x$ (defined in Fig. 4e) and energy, with the gap indicated by two horizontal dashed lines. A red dashed rectangle in Fig. 4g highlights the enhanced edge conductance within the gap near the island edge. Note that STS can be affected by quasiparticle interference (QPI) effects, as indicated in Fig. 4g; similar effects have been seen in WTe$_2$[29]. For curve A in Fig. 4f, the strong peak at about −170 meV, which might seem strange, is caused

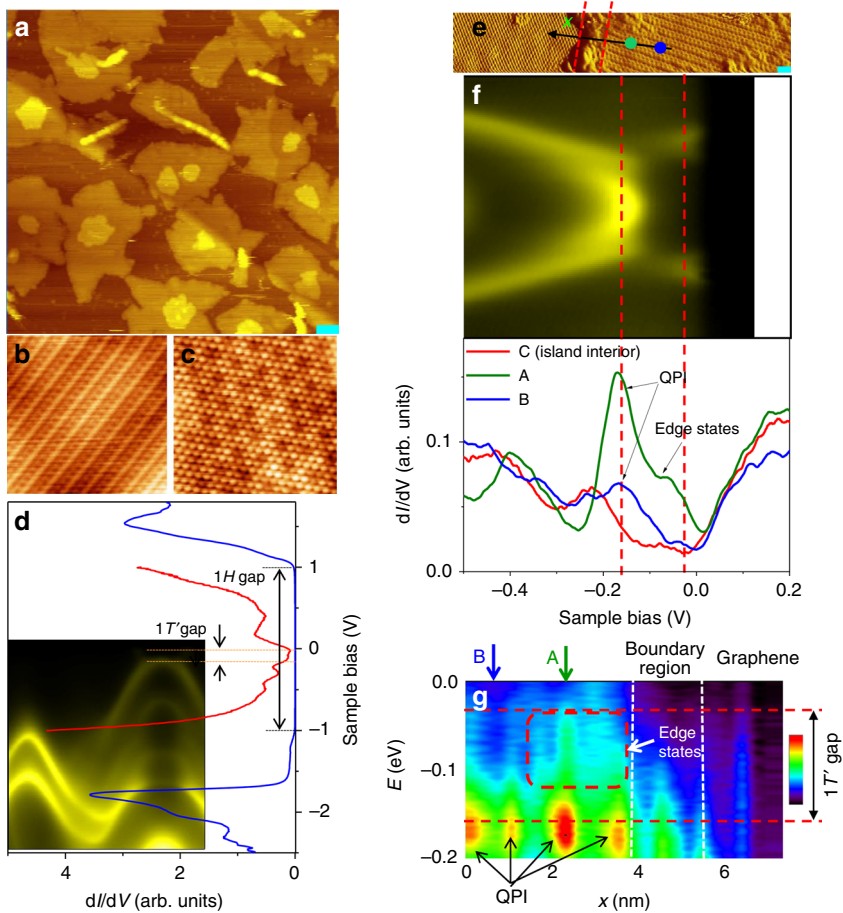

**Fig. 4** STM/STS data. **a** Topographic image of a sample with a nominal 1/2-layer coverage showing mixed 1T′ and 1H phases. Image size: 150 nm × 145 nm; $V_s = 1.43$ V; $I_t = 0.35$ nA. The scale bar is 10 nm. **b** An atomic image over a 1T′ region (6.0 nm × 6.0 nm; $V_s = -1.0$ V; $I_t = 0.77$ nA). **c** An atomic image with Moiré modulation over a 1H region (5.5 nm × 5.5 nm; $V_s = -1.0$ V; $I_t = 0.77$ nA). **d** STS spectra taken at the interior of an 1H island (blue curve) and a 1T′ island (red curve), overlaid onto an ARPES map; the corresponding gaps are indicated. The intensity of the red curve is amplified six times for clarity. **e** An image showing a 1T′ island on the right with a scale bar of 1 nm. **f** Color-coded STS spectra taken at a point very close to an island edge (curve A), a point still near but farther away from the edge (curve B), and another point deep inside the island (curve C). Points A and B are indicated by the correspondingly color-coded dots in **e**. Curve C shows a gap with low background intensity within the gap. Curve A shows a high intensity within the gap, consistent with the presence of edge states; this effect is much reduced for curve B. The pronounced peak for curve A at about −170 meV arises from constructive QPI. **g** A 2D STS map as a function of x (defined in **e**) and energy. The gap is indicated by two horizontal lines, and a region with strong edge-state intensity is highlighted by a dashed rectangle. QPI oscillations near the valence band top are indicted. All data were taken at 77 K, and STS was conducted with a 7.5 mV modulation at 5 kHz

by QPI constructive interference (the strongest red spot indicated in Fig. 4g). Despite this complication, the edge-state contribution within the gap is evident. The results are consistent with the earlier conclusion of a QSH system. However, observation of edge conductance alone does not prove the QSH nature of the system; such conductance could arise from other, non-topological edge states.

## Discussion

To design or search for materials with large QSH gaps, one might be tempted to employ chemical substitution of the constituent atoms of a QSH system with heavier elements in the same column of the periodic table. Our observation of a QSH gap in single-layer 1T′ WSe₂ about twice as large as that in WTe₂ seems counter-intuitive. As discussed above, the QSH gap in WSe₂ originates from band inversion of the W 5d and Se 4p states near the zone center and anti-crossing of the bands caused by

spin–orbit coupling (Fig. 2c). A similar theoretical analysis for WTe₂ shows that the relevant reverse-ordered states near the zone center are both dominated by the W 5d states[30]. The Te states play a relatively minor role. The self-hybridization of the W 5d states is actually weaker, leading to a smaller QSH gap.

Most single-layer MX₂ materials, including WSe₂, are stable only in the 1H phase, which gives rise to ordinary semiconductors. Our demonstration of the successful creation of single-layer 1T′ WSe₂ with a sizable QSH gap offers an important example of materials engineering. Its QSH gap of 129 meV is more than five times the thermal energy $k_B T$ at room temperature, suggesting that it is suitable for QSH electronics at ambient temperature. While the system is only metastable, its demonstrated stability up to ~280 °C is more than sufficient. For comparison, 1T′ WTe₂, which has garnered much attention, actually has a smaller QSH gap not conducive to ambient-temperature spintronics. Unlike exfoliated materials, the 1T′ WSe₂ films grown by molecular beam epitaxy, as demonstrated herein,

should be readily adaptable to large-scale fabrication of devices such as topological field effect transistors[16]. Other materials such as superconductors can be added by molecular beam epitaxy, thus offering opportunities to realize additional functionality and novel properties including Majorana fermions. Our work expands the family of large-gap QSH materials and inspires further experimental exploration of novel QSH systems.

## Methods

**Experimental details.** Thin films of $WSe_2$ were grown in situ in the integrated molecular beam epitaxy (MBE)/ARPES systems at beamlines 12.0.1 and 10.0.1 (Advanced Light Sources, Lawrence Berkeley National Laboratory). Substrates of 6H-SiC(0001) were flash-annealed for multiple cycles to form a well-ordered bilayer graphene on the surface[23]. Films of $WSe_2$ were grown on top of the substrate at a rate of 50 min per layer by co-evaporating W and Se from an electron-beam evaporator and a Knudsen effusion cell, respectively. The growth of different phases of $WSe_2$ is controlled by the substrate temperature. The $1T'$ phase starts to form at 130 °C but the film quality is poor at low growth temperatures. The best $1T'$ phase is obtained at near 280 °C, but the $1H$ phase also forms and completely dominates at temperatures at about 400 °C. ARPES measurements were performed with an energy resolution of <20 meV and an angular resolution of 0.2°. Each sample's crystallographic orientation was precisely determined from the symmetry of constant-energy-contour ARPES maps. The surface electron density with Rb doping is determined from the Luttinger area of the Fermi surface around $\bar{\Gamma}$ based on $N = g_v \frac{k_F^2}{2\pi}$[22]. For STM/STS measurements, the samples after characterization by ARPES were capped with a 20 nm layer of Se for protection and then shipped and loaded several days later into the STM/STS system, wherein the protective Se layer was thermally desorbed at 250 °C before measurements using an Omicron LTSTM (low-temperature scanning tunneling microscopy) instrument and freshly flashed tungsten tips. Experimentation with capping and decapping using the ARPES system reveals that such sample treatments lead to a clean but rougher surface, as evidenced by decreased clarity of the ARPES maps but no detectable impurity core level peaks.

**Computational details.** First-principles calculations were performed using the Vienna ab initio package (VASP)[31–33] with the projector augmented-wave method[34,35]. A plane wave energy cut-off of 400 eV and an $8 \times 8 \times 1$ $k$-mesh were employed. The GGA with the PBE functional[36] was used for structural optimization of single-layer $WSe_2$. Freestanding films were modeled with a 15 Å vacuum gap between adjacent layers in the supercell. The fully optimized in-plane lattice constants for single-layer $1T'$ $WSe_2$ are $a = 5.94$ Å and $b = 3.30$ Å. Our results generally agree with those reported in ref. [15] where available.

**Data availability.** The data that support the findings of this study are available within the article or from the corresponding author upon request.

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

## Acknowledgements

This work is supported by the US Department of Energy, Office of Science, Office of Basic Energy Sciences, Division of Materials Science and Engineering, under Grant No. DE-FG02-07ER46383 (to T.-C.C.), the National Science Foundation under Grant No. EFMA-1542747 (to M.Y.C.), and the Ministry of Science and Technology of Taiwan under Grant No. 104-2112-M-002-013-MY3 and the Center of Atomic Initiative for New Materials, National Taiwan University (to W.-P.). The Advanced Light Source is supported by the Director, Office of Science, Office of Basic Energy Sciences, of the US Department of Energy under Contract No. DE-AC02-05CH11231. The work at Academia Sinica is supported by a Thematic Project.

## Author contributions

P.C. and T.-C.C. designed the project. P.C. with the aid of C.-Z.X., A.-V.F., and T.-C.C. performed MBE growth, ARPES measurements, and data analysis. Y.-H.C. and M.Y.C. performed first-principles calculations. W.-W.P., W.-L.S., and D.-S.L. conducted STM/STS experiments. T.-C.C., P.C., W.-W.P., and M.Y.C. interpreted the data. T.-C.C. and A.-V.F. jointly led the ARPES project. P.C. and T.-C.C. wrote the paper with input from other co-authors.

## Additional information

**Competing interests:** The authors declare no competing interests.

