## [Peer Review File · Nature Communications]

Reviewers' comments:

Reviewer #1 (Remarks to the Author):

I reviewed this paper previously. I am mostly satisfied with the authors' response to my comments, and I now recommend publication of the revised MS in its present form.

Reviewer #2 (Remarks to the Author):

I think the paper is improved but I still think the authors are putting too much weight on features from STM spectra. I take strong exception to starting a paragraph with 'Experimental observation of edge states within the gap is a necessary test for QSH systems'..and then discussing the STM spectra as though they are proof of the existence of topological 1D modes.

The STM data are certainly not proof (as the authors correctly point out). However, there are so many explanations for what they see that they are not even 'consistent'. The STM data is extremely sketchy. Point spectra can be picked to show anything. The authors show only three spectra. This is not sufficient for any analysis what so ever.

What the authors would need for 'consistency' is multiple data sets with multiple tips showing:

- 1) Everywhere in the interior of the island there is a gap.
- 2) Everywhere on the edge there is a mode at a particular energy.
- 3) A calculation showing what you expect to see.

This is however a lot of work and I don't expect the authors to do this for this paper.

Instead, what I would like to see is for the authors to NOT claim to be seeing edge modes based on the STM spectra. This is already a very nice paper without this.

Simply use the STM to show the existence of the two types of structures and stop at that. They can show the spectra if they wish of course but it would be extremely disingenuous to draw any generalizations based just on a few point spectra (or even a spatially averaged spectrum over a small region).

The other problem with the STM spectra is the QPI peak. I am not sure what this means and on what basis the authors are choosing to call this a QPI peak. I would not label the peak as such.

In summary: the STM spectroscopy data and discussion is not at all robust and even unnecessary. Its scientifically unsound to base anything on the limited data they have. I would strongly recommend another revision where this data and discussion are rewritten to take these comments into account. Without this, I would not recommend publication.

Authors' point-by-point response to reviewers' comments

Reviewer #1 (Remarks to the Author):

In this work the authors report the growth of WSe₂ single layer with the 1T' structure that does not exist in the bulk. ARPES studies show a gap of 129 meV in the 1T' layer. The observed gap decreases with doping by Rb adsorption. They further confirm the 2D TI character by first-principles calculations. They also carry out STM studies on the sample. The ARPES studies are robust and clearly show a well defined gap for the 1T' phase. The STM studies are however not so robust.

Authors' response: We thank the reviewer for finding our ARPES results robust and our theoretical analysis satisfactory. The gap structure is made clear by Rb doping in order to bring the conduction bands into view by ARPES. STM was used to confirm the 1T' structure which is distinct from the 1H structure. STS data, as supporting information, were included to show consistency with the QSH behavior. We appreciate the referee's concerns about the STS part, and we apologize for not being very clear in the original write-up with the data presentation and labeling in Fig. 4. We have fixed the issues as explained below. For easy reference, the revised Fig. 4 as well as a list of major changes is included at the end of this document.

First, the STM spectra do not show a full gap for the 1T' phase. The question then arises: where is the density of states coming from?

Authors' response: In the original Fig. 4f, we showed two STS curves taken at a point (A) very close to an island edge (curve A) and another point (B) farther away from the edge (curve B). Curve A shows substantial tunneling density of states (DOS) within the gap due to the edge states. Curve B shows much less DOS, but it is still significant, as noted by the referee as a concern for not seeing a full gap; however, point B is not far enough away from the island edge. In the revised Fig. 4f, we have added another STS curve (curve C) taken from a point deep inside the island. This reference curve shows a clear gap; the small residual DOS within the gap can be attributed to tunneling coupling to the underlying bilayer graphene. Similar nonzero DOS in the gap is evident in results taken from single-layer WTe₂ [Ref. 17]. The revised Fig. 4f also contains an ARPES map to show the gap location. Figure 4g has been revised to show the locations of the two points A and B relative to the island edge where the STS data were taken. Additional labels have been added to indicate the relevant features. The discussion in the text has been correspondingly updated.

Second, the extra states at the edge are not necessarily topological. Many islands show a pile up of density of states on the edges even without non-trivial topology.

Authors' response: We agree that STS of the edge states does not provide topological information directly. Our work is primarily an ARPES study of the band structure of the 1T' phase to accurately map out the gap structure. The ARPES results coupled with first-principles calculations establish a large quantum spin Hall (QSH) gap. The edge states, while "topologically protected", can still be influenced in detail by the exact structure of the edge. Our theoretical analysis of the band characters confirms that the topological index $Z_2 = 1$ for single-

layer 1T' WSe₂ in agreement with a prior theoretical study [Ref. 16], and thus this is indeed a QSH system. Our STS data in Fig. 4f revealing edge states within the band gap are consistent with the QSH behavior but do not offer an independent proof.

All in all, the ARPES data is interesting. It shows a gap that suggests that the material may be a candidate QSH system. The change in band structure with doping suggests interesting correlated physics.

The STM studies however are far from convincing. Moreover the quality of the STM data is not high and the conclusions are hasty. I would not recommend publication in its current form.

Authors' response: We thank the reviewer's comment that our ARPES data are interesting. Our work is primarily an ARPES study of the band structure of a novel 1T' phase, rather than the usual 1H phase, of WSe₂ created by a careful growth experiment. The ARPES results coupled with first-principles calculations establish a large quantum spin Hall (QSH) gap (twice as large as that in WTe₂, which has been at the center of attention of the community). The larger gap in WSe₂ is unexpected based on conventional wisdom but is now understood from our theoretical analysis. The larger gap makes the system a prime candidate for room temperature spintronic operations, which is of great technological interest. In our work, STM was used to confirm the 1T' structure that is distinct from the 1H structure. STS data were included to show consistency with the QSH behavior. We agree that STS alone cannot address the topological property directly, but this information is already established by ARPES and band calculations. The revised Figs. 4f and 4g and associated discussion in the text should be much more informative than before regarding the gap location and residual density of states. The revision also includes a discussion of quasiparticle interference (QPI) effects that affect the STS spectra as labeled in Fig. 4g. Note that similar QPI effects have been reported for WTe₂ [Ref. 29].

Reviewer #2 (Remarks to the Author):

In this manuscript, Chen et al. reported a large-gap quantum spin Hall insulator state in single-layer 1T' WSe₂. Below are my review comments.

(1) In terms of significance and general interest, the quantum spin Hall (QSH) states in 1T' MX₂ (M=Mo, W; X=S, Se, Te) has attracted much interest with several experimental reports [Nature Phys. 13 677 (2017); Nature Phys. 13, 683 (2017); PRB 96, 041108(R) (2017), and most recently Science 359, 76–79 (2018)] after original theoretical prediction [Science 246, 1345 (2014)]. So the experimental realization of the QSH state in the metastable 1T' WSe₂ is no longer surprising. However, I think the reported large band gap of 1T' WSe₂, twice as large as that of 1T' WTe₂, is an important finding, provided its topological nature is well established as I commented below.

Authors' response: We thank the reviewer for finding our work important.

(2) My main concern is the experimental data presented in Fig. 4, which to me is not convincing or somewhat confusing. In fig. 4b, one sees the bulk gap of 1T' phase in the energy range from 0 to about -139 meV. However, there was no edge states in this same energy range in Fig. 4f and 4g. Instead one see peaks around -175 meV. Even more troublesome is that one sees the peak at this energy in both bulk and edge spectra (Fig. 4f and 4g) albeit larger intensity in the edge spectra. In essence, the authors have to establish the bulk-edge correspondence to confirm the topological edge state rather than trivial chemical edge state. This could be done by a carefully mapping of measured edge state to the DFT calculated edge state as previously reported in the literature for very similar topological edge-state studies [see, e.g., Phys. Rev. Lett. 109, 016801 (2012); Nature Mat., 15, 968 (2016)]. Maybe I am confused by their presentation, but it is not clear and convincing to me.

Authors' response: Figure 4 has been revised (included at the end of this document together with a list of major changes). We apologize for not being very clear in the original write-up with the data presentation and labeling in Figs. 4f and 4g. We have fixed the issues in the revised version. In the original Fig. 4f, we showed two STS curves taken at a point (A) very close to an island edge (curve A) and another point (B) farther away from the edge (curve B). Curve A shows substantial tunneling density of states (DOS) within the gap due to the edge states. Curve B shows much less DOS, but it is still significant; however, point B is not far enough away from the island edge. In the revised Fig. 4f, we have added another STS curve (curve C) taken from a point deep inside the island. This reference curve shows a clear gap; the small residual DOS within the gap can be attributed to tunneling coupling to the underlying bilayer graphene. Similar nonzero DOS in the gap is evident in results taken from single-layer WTe₂ [Ref. 17]. The revised Fig. 4f also contains an ARPES map to show the gap location. Figure 4g has been revised to show the locations of the two points A and B relative to the island edge where the STS data were taken. Additional labels have been added to indicate the relevant features.

Please note that STS near an island edge can be affected by quasiparticle interference (QPI) effects. For curve A in Fig. 4f, the big peak at about -175 meV noted by the referee is actually a QPI peak. Such QPI effects are absent for curve C taken from a point very far away from island edges, and are much reduced for curve B. It is perhaps best to examine the overview provided by

the revised Fig. 4g, which shows a map of the tunneling spectra near the edge. The gap is now marked by two horizontal lines. Somewhat below the valence band top (the lower horizontal dashed red line) are strong QPI oscillations as a function of distance from the island edge arising from the fairly flat portion of the valence band near its maximum (Fig. 2). This is the reason for the strong peak at about -175 meV noted earlier. We have added a dashed rectangle in Fig. 4g to highlight the enhanced DOS from edge states within the gap; this extra DOS is located very close to the island edge. Note that similar QPI effects have been reported for WTe_2 [Ref. 29]. The discussion in the text has been updated correspondingly.

(3) The authors pointed out that their finding of a larger gap in 1T' WSe_2 than in WTe_2 is counter-intuitive, agreed. But their explanation is not convincing enough. It is known that the band-inversion mechanisms in 1T' WSe_2 (p-d inversion) and 1T' WTe_2 (d-d inversion) are different [PRB 93, 125109 (2016)]. Then one would expect that the SOC induced QSH gap should be larger in WTe_2 since the atomic SOC strength of d orbitals is larger. The authors mentioned that the reverse-ordered states near the zone center are both derived from the W 5d state and the Te state does not come into play. However, since WTe_2 is a strong covalent compound, the orbitals of W and Te should hybridize to form bands. Actually, the orbital contribution of 1T' WTe_2 shown in previous works [supplemental material of Nature Phys. 13, 683 (2017)] indicate that both Te p and W d orbitals contribute to the bands around the Fermi level.

Authors' response: We thank the reviewer's insights. The wave functions do involve a mixture, and the mixing depends on the wave vector. We have added the reference PRB 93, 125109 (2016) and revised the related text. The revised text now reads "A similar theoretical analysis for WTe_2 shows that the relevant reverse-ordered states near the zone center are both dominated by the W 5d states. The Te states play a relatively minor role. The self-hybridization of the W 5d states is actually weaker, leading to a smaller QSH gap."

(4) In the introduction when citing previous works on 2D elemental TI with a large gap, the original theoretical proposal to grow 2D metal overlayer (such as Bi, Pb etc) on patterned semiconductor substrates (such as Si and SiC) (PNAS 111, 14378 (2014)), which has been experimentally confirmed recently [i.e., Ref. 14, Science 357, 287 92017]], should be cited along with Refs. 8-14.

Authors' response: We thank the reviewer's suggestion. This paper has been added as Ref. 15.

In summary, I do not recommend publication of this MS in Nature Nanotechnology in its present form. However, I may reconsider if the authors can fully address my above-mentioned concerns, especially point #2.

Authors' response: Thank you very much for the consideration. We hope that you will be pleased with the revision.

Summary of major changes to the manuscript

1. **Figure 4 has been improved and reorganized. A copy is shown below.**
2. **A paragraph associated with Fig. 4 has been updated. It is included below.**
3. Caption for Fig. 4 has been updated. Please see revised manuscript.
4. References suggested by the referees have been added. Please see revised manuscript.

1. Revised Fig. 4:

2. Revised paragraph associated with Fig. 4:

Experimental observation of edge states within the gap is a necessary test for QSH systems. Figure 4e is an STM image with a 1T' island covering the right half only. Figure 4f shows STS curves taken at a point (A) in the island very close to the island edge (green curve A) and another point (B) in the island still near but farther away from the edge (blue curve B); the two points A and B are indicated by the correspondingly color-coded dots in Fig. 4e. With the Fermi level aligned relative to the included ARPES map²⁸, the gap is indicated by the two vertical red dashed lines. Also shown is a reference red curve C taken from a point deep inside the island. It shows a clear gap; the small residual tunneling density of states (DOS) within the gap can be attributed to tunneling/coupling to the underlying bilayer graphene. Similar nonzero DOS in the gap is evident in single-layer WTe₂ results¹⁷. Curve A, relative to curve C, shows much higher DOS within the gap, suggesting contributions from edge states^{13, 14, 17}, a key feature of QSH systems; this extra DOS is much reduced for curve B. Details regarding the edge-state contributions are shown in Fig. 4g, where the differential conductance is plotted as a function of x (defined in Fig. 4e) and energy, with the gap indicated by two horizontal dashed lines. A red dashed rectangle in Fig. 4g highlights the enhanced edge conductance within the gap near the island edge. Note that STS can be affected by quasiparticle interference (QPI) effects, as indicated in Fig. 4g; similar effects have been seen in WTe₂²⁹. For curve A in Fig. 4f, the strong peak at about -170 meV, which might seem strange, is caused by QPI constructive interference (the strongest red spot indicated in Fig. 4g). Despite this complication, the edge-state contribution within the gap is evident. The results are consistent with the earlier conclusion of a QSH system.